# Evaluating the documentation of vital signs following implementation of a new comprehensive newborn monitoring chart in 19 hospitals in Kenya: A time series analysis

**Naomi Muinga** [1,2,3] *, **Timothy Tuti** [1], **Paul Mwaniki** [1], **Edith Gicheha** [4], **Chris Paton** [5,6], **Lenka Beňová** [3], **Mike English** [1,5]

**1** Athena Institute, VU University Amsterdam, Amsterdam, Netherlands, **2** KEMRI/Wellcome Trust Research Programme, Nairobi, Kenya, **3** Department of Public Health, Institute of Tropical Medicine, Sexual and Reproductive Health Group, Antwerp, Belgium, **4** Rice360 Global Health Institute, Rice University, Texas, United States of America, **5** Nuffield Department of Medicine, Health systems Collaborative, University of Oxford, Oxford, England, **6** Department of Information Science, University of Otago, Dunedin, New Zealand

* nmuinga@kemri-wellcome.org

**Data Availability Statement:** The datasets generated and/or analysed during the current study

## Abstract

Multi-professional teams care for sick newborns, but nurses are the primary caregivers, making nursing care documentation essential for delivering high-quality care, fostering teamwork, and improving patient outcomes. We report on an evaluation of vital signs documentation following implementation of the comprehensive newborn monitoring chart using interrupted time series analysis and a review of filled charts. We collected post-admission vital signs (Temperature (T), Pulse (P), Respiratory Rate (R) and Oxygen Saturation (S)) documentation frequencies of 43,719 newborns with a length of stay > 48 hours from 19 public hospitals in Kenya between September 2019 and October 2021. The primary outcome was an ordinal categorical variable (no monitoring, monitoring 1 to 3 times, 4 to 7 times and 8 or more times) based on the number of complete sets of TPRS. Descriptive analyses explored documentation of at least one T, P, R and S. The percentage of patients in the no-monitoring category decreased from 68.5% to 43.5% in the post-intervention period for TPRS monitoring. The intervention increased the odds of being in a higher TPRS monitoring category by 4.8 times (p<0.001) and increased the odds of higher monitoring frequency for each vital sign, with S recording the highest odds. Sicker babies were likely to have vital signs documented in a higher monitoring category and being in the NEST360 program increased the odds of frequent vital signs documentation. However, by the end of the intervention period, nearly half of the newborns did not have a single full set of TPRS documented and there was heterogenous hospital performance. A review of 84 charts showed variable documentation, with only one chart being completed as designed. Vital signs documentation fell below standards despite increased documentation odds. More sustained interventions are required to realise the benefits of the chart and hospital-specific performance data may help customise interventions.

are not publicly available due to the primary data being owned by the hospitals and their counties with the Ministry of Health; The research staff do have permission to share the data without further written approval from both the KEMRI-Wellcome Trust Data Governance Committee and the Facility, County or Ministry of Health as appropriate to the data request. Requests for access to primary data from quantitative research by people other than the investigators will be submitted to the KEMRI-Wellcome Trust Research Programme data governance committee as a first step through dgc@kemri-wellcome.org who will advise on the need for additional ethical review by the KEMRI Research Ethics Committee.

**Funding:** NM was supported through the DELTAS Africa Initiative [DEL-15-003]. The DELTAS Africa Initiative is an independent funding scheme of the African Academy of Sciences (AAS)'s Alliance for Accelerating Excellence in Science in Africa (AESA) and supported by the New Partnership for Africa's Development Planning and Coordinating Agency (NEPAD Agency) with funding from the Wellcome Trust [107769/Z/10/Z] and the UK government. Funds from a Wellcome Trust Senior Clinical Research Fellowship (#207522) awarded to Professor Mike English supported this work together with a core grant awarded to the KEMRI-Wellcome Trust Research Programme (#092654) and a grant to the NEST program from the John D. and Catherine T. MacArthur Foundation, the Bill & Melinda Gates Foundation, ELMA Philanthropies, and The Children's Investment Fund Foundation UK under agreements to William Marsh Rice University with a sub-agreement through the University of Oxford Centre for Tropical Medicine and Global Health. The views expressed in this publication are those of the author(s) and not necessarily those of AAS, NEPAD Agency, Wellcome Trust, the UK government or other funders. The funders had no role in study design, data collection and analysis, the decision to publish, or preparation of the manuscript.

**Competing interests:** The authors have declared that no competing interests exist.

# Introduction

Nurses are the primary caregivers in newborn units and documentation of nursing care facilitates the provision of individualised quality care and fosters team communication to improve patient outcomes [1]. Care for sick newborns, who spend many days in the neonatal unit, is typically provided by multi-professional teams [2]. This makes documentation of care an essential activity as it provides a common reference point in knowing what care has or has not been provided as well as tracking the newborn's progress. Common causes of newborn mortality include preterm birth complications, birth asphyxia, and neonatal sepsis [3] and essential interventions have been identified to reduce preventable newborn deaths in hospitals [4,5]. The provision of quality care is an essential step toward reaching the Sustainable Development Goal (SDG) 3.2 target of reducing neonatal mortality to below 12 per 1000 live births [6].

Many low- and middle-income countries (LMICs) have nursing staff shortages and this problem is particularly acute in neonatal inpatient units where babies need round-the-clock nursing care. Information on patient status and current and planned care must be clearly documented to coordinate the care team and recognise patient deterioration. To facilitate this, job aids such as rounding checklists [7], structured clinical forms [8] and forms guiding the handover of patients between staff shifts [9] are used to help structure and standardise work processes and contribute to making them safer and easier.

In prior work in Kenya, we found that vital signs charts used for inpatient monitoring were commonly available but were not specifically designed for newborns and were completed for only a third of sick newborns [10]. In subsequent work to define contextually appropriate standards for neonatal nursing care, it was proposed that better tools to document nursing observations were needed to facilitate more rapid, accurate, and informative communication between nurses and other professionals to improve the quality of care [11]. To meet this need, we used a human-centred design approach to develop a new paper-based nursing monitoring chart for sick newborns, with information on this process provided elsewhere [12].

Here we report the results from an intervention study in which the newly designed Comprehensive Newborn Monitoring (CNM) chart was introduced into Clinical Information Network hospitals' (CIN-N) newborn units in Kenya [13]. Due to the COVID-19 pandemic, the implementation was conducted virtually in July 2020 through an online meeting attended by nurses.

This study aimed to quantitatively evaluate the recording of vital signs (Temperature, Pulse, Respiratory Rate, and Oxygen Saturation) following the implementation of the newly designed CNM chart. We also reviewed filled charts to supplement the quantitative evaluation.

# Methods

This is a quantitative study comprising a quantitative evaluation of an intervention (introduction of CNM charts) using routine data collected in the CIN-N database over 24 months (September 2019 to October 2021). This was supplemented by an explanatory review of the quality of documentation of completed charts.

## Study setting

The CIN-N is a partnership between researchers (and indirectly their funders), the Kenyan Ministry of Health, the Kenya Paediatric Association and county hospitals. It was established in 2013 and has expanded over time. In 2018, a database that collects routine data from newborn wards, CIN-N, was initiated and now constitutes 22 public hospitals. The CIN-N aims to improve the use of information for both policymaking at the national level and decision making at the hospital level and, therefore, to improve the outcomes of the babies admitted to the

hospitals. A trained data clerk abstracted after discharge from hospital records into a custom-ised database data through an informatics framework that is described in detail elsewhere [14,15]. Since 2018, quarterly feedback reports are sent to all hospitals, as are monthly mortal-ity reports that include feedback on documentation quality, including, among other items, indicators for documentation of vital signs on admission [14].

## The intervention

The CNM chart has a large section that contains 12 columns for documenting repeated mea-sures of vital signs and other assessments and it can be used flexibly as there are no fixed tim-ings in the top row. The chart is designed to cover 48 hours if printed double-sided on A4 paper. This design maximises the use of paper in a low-resource setting while allowing for comprehensive monitoring of vital signs, feed, and fluid documentation on a single sheet. The nursing team fills all sections except the feed and fluid prescription section, which is filled by the clinicians [12].

A virtual chart launch was conducted in July 2020 and the hospitals received a distribu-tion package that included a two-month supply of monitoring charts in August 2020 [13]. The intervention start month was September 2020(month 0) and data from 19 public hos-pitals in the CIN-N network with pre- and post-intervention data from across Kenya were used for these analyses (S1 Appendix). Three of the hospitals adopted the chart earlier than the expected chart launch date and are referred to as Early Adopter (EA) hospitals (Table 1).

## Landscape of other projects

The nature of CIN-N is such that it provides a framework onto which interventions can be implemented in participating hospitals and patient outcomes and hospital performance tracked using routine data [16]. Importantly, CIN-N in general provides no additional clini-cal or nursing staff or other material resources for Newborn Units. However, some CIN-N hospitals are involved in additional projects or programmes potentially affecting newborn care practices. These are the Newborn Essential Services and Technologies (NEST360) Pro-gram [17] which provides key medical equipment, clinical and biomedical engineering training, supportive supervision including 3 monthly Quality Improvement visits, and a smaller project that involved designing, testing and implementing newborn clinical audit tools. In brief, CIN-N activities and complementary projects during the intervention period are listed below and details of the specific projects that hospitals were involved in are pro-vided in Table 1.

- NEST360 program– 12 hospitals received essential newborn equipment including pulse oximeters and staff were trained and mentored on comprehensive newborn care including its documentation [17].

- 4 hospitals were involved in the codesign of team-based small and sick newborn clinical audit tools [18]–these hospitals were also part of the NEST360 program

- 14 hospitals in which at least one senior nurse was involved in communication and emo-tional competence training [19]

- 15 hospitals were involved in the co-design of monitoring charts (the intervention assessed in this study) [12],

Additionally, during the CNM chart implementation period, the NEST360 program con-ducted a webinar on monitoring and documenting newborn care (April 29, 2021) and all

**Table 1. Complementary projects start dates and network activities.**

| | Project | |
| --- | --- | --- |
| Hospital code | Co-design of CNM chart | NEST (joined in month-year) |
| H1 | Y | Oct-20 |
| H2 | | |
| H3 | Y | Apr-21 |
| H4 | | |
| H5 | Y | Apr-21 |
| H6 | Y | Oct-20 |
| H7 | Y | Nov-20 |
| H8 | Y | Mar-21 |
| H9 | Y | |
| H10 (EA) | Y | Nov-20 |
| H11 (EA) | Y | Feb-20 |
| H12 | Y | |
| H13 | | |
| H14 | Y | Mar-21 |
| H15 (EA) | Y | Jan-20 |
| H16 | Y | Feb-21 |
| H17 | Y | Nov-20 |
| H18 | | |
| H19 | | |

Network Activities (All hospitals).

1. Monthly Mortality reports that included feedback on admission documentation quality.

2. Quarterly Morbidity Reports.

3. Webinar on newborn monitoring facilitated by NEST360 program April 2021.

*Blank spaces mean the hospital was not involved.

EA–Early adopter hospitals.

health workers (from CIN-N hospitals and non-network hospitals) were invited. It was conducted as part of the NEST360 mentorship efforts to strengthen newborn care in response to an observation of poor monitoring trends in CIN-N hospitals.

## Study population, data collection

From a recent study, the crude in-facility newborn mortality rate in CIN-N hospitals was 10.2% (95% CI 9.97% to 10.55%), with 60% of these deaths occurring within 24 hours of admission [20]. All patients in the newborn wards admitted to 19 hospitals on the day of birth, with a length of stay (LOS) > 48 hours and with a full dataset (detailed clinical data collected upon discharge [15] were eligible for inclusion in the analysis. In some especially busy hospitals/periods only 'minimum data' that did not include vital signs documentation were collected from a random selection of discharged patients and thus their data were excluded. A length of stay (LOS) > 48 hours was used as an inclusion criterion to ensure a standard period over which the number of vital signs' observations documented was determined. Data from all 19 hospitals are used for descriptive analyses. Three hospitals opted to adopt the CNM chart before its official launch (EAs; all were involved in chart co-design) and were excluded from our Interrupted Time Series (ITS) analysis, as they were likely to mask any intervention effect. Fig 1 shows data used for analyses.

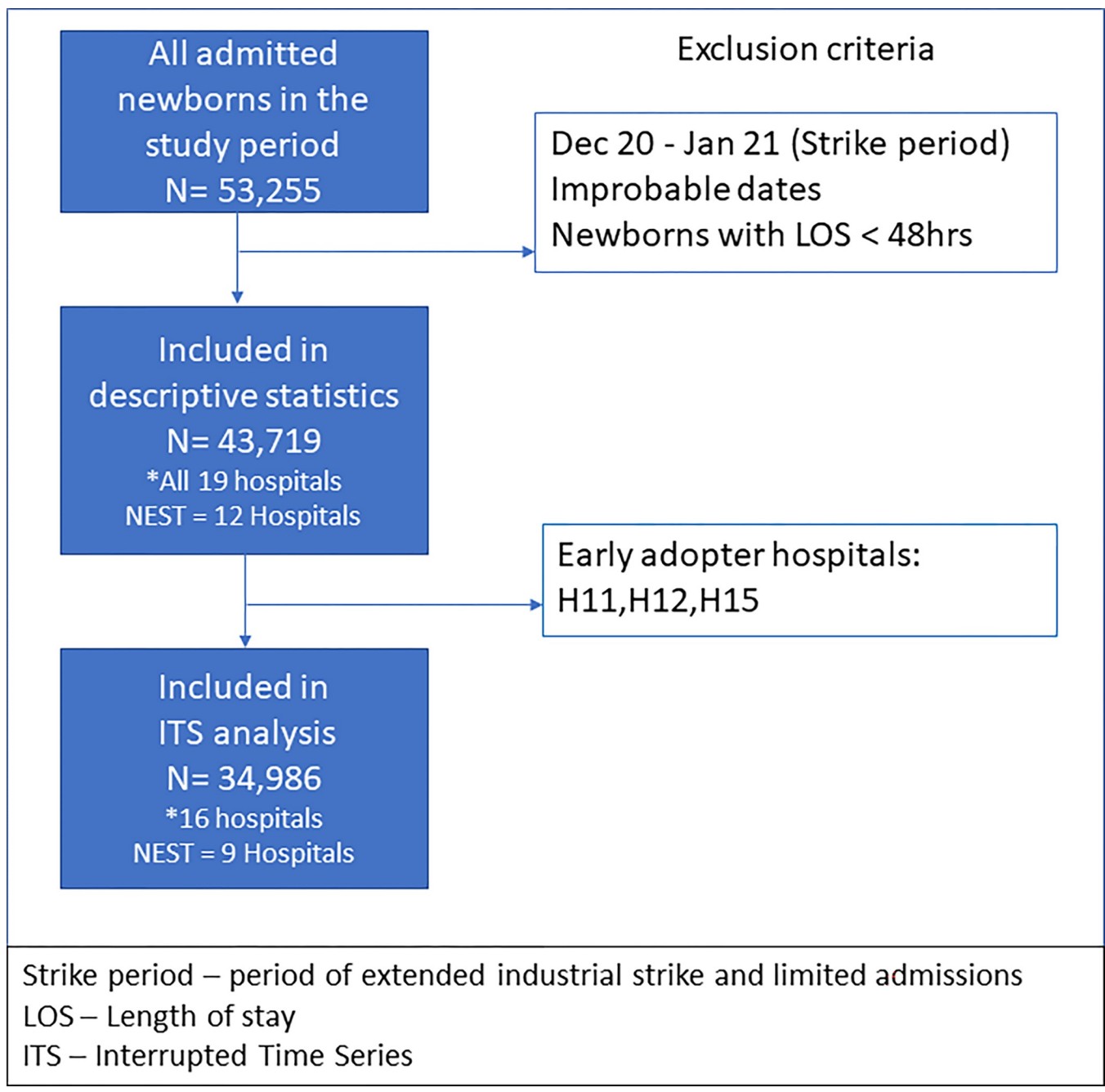

**Fig 1. Available data.**

### Intervention evaluation design

**Main outcome–TPRS composite outcome.**   To explore the effect of CNM chart introduction, we used an ITS analysis, which relies on pre- and post-intervention data and complement this with an audit of hospitals' monitoring charts. Previous work has shown that it can take up to 6 months for interventions to stabilise [21] and therefore, periods of 12 months pre- and post-intervention would give a fair representation of any changes in documentation trends. Randomisation of hospitals to intervention was not feasible. The primary reason was that

charts were co-designed with the hospital staff, making it ethically difficult to deny or postpone hospitals' access to a chart that they helped develop [12]–this was supported by the emergence of early adopter hospitals (Table 1).

Vital signs monitoring standards had previously been agreed upon with the Kenyan nursing community [11]. For sick babies admitted to newborn units it is recommended that Temperature (T), Pulse (P), Respiratory Rate (R) and Oxygen Saturation (S) measurements are performed and documented, at each monitoring time point with the frequency determined by the severity of illness. Data were collected as the number of times each vital sign was monitored in the first 48 hours of admission (with anything above 10 coded as = >10). It was anticipated that the vital signs readings would be taken and documented together during each monitoring episode. To confirm this, we tested for correlation between the number of times specific vital signs were documented. Frequency of T, P and R documentation were very highly correlated (Pearson's correlation $\rho>0.9$) while S (oxygen saturation or cyanosis assessment) was slightly less highly correlated with the other vital signs (Pearson's correlation $\rho>0.8$). Therefore, we created an ordinal outcome with four categories based on the number of times a complete set of the four vital signs was documented as illustrated in Table 2. For example, if upon discharge, the vital signs were documented as 2T, 2P, 2R, but 1S, then this individual is considered having 1 complete set of TPRS observations.

## Data analysis

Data analysis was conducted in three steps. First was a descriptive analysis that was conducted on two levels (with and without the three EA hospitals). This was followed by an ITS analysis which excluded EA hospitals. Last, an exploratory data quality a review of a sample of filled monitoring charts to complement the quantitative data analysis was conducted.

**Descriptive analysis.** In the descriptive analysis, observed percentages of babies in the four main outcome categories based on the documented vital signs in the two periods (before and after the CNM chart launch) are presented using bar charts. Both bar charts and line charts are presented for all hospitals combined with and without early adopters. Additionally, the findings for individual hospitals are provided in Figs 1 and 2 in S2 Appendix.

**Explanatory analyses–individual vital signs.** The vital signs were also analysed individually to understand how recording practices for each vital sign changed over time. Line graphs showing the percentage of newborns with at least one recording of a specific vital sign (T, P, R, or S) in the 48 hours post-admission are presented over the study period.

**Interrupted Time Series (ITS) analysis–model specification.** The ITS analysis modelled the TPRS composite as ordinal categorical outcomes. A multilevel segmented ordinal logistic regression model was applied to account for the pre- and post-intervention trends in the study outcomes (both composite TPRS and individual vital signs recoded into the categorical outcome) [22]. Informed by previous findings [5] and our previous experience in the CIN-N, our

**Table 2. Categorical outcomes for vital signs monitoring in the first 48-hour period after admission.**

| Number of times monitored | Name of category | Description |
|---|---|---|
| 0 | No Monitoring 0 | Post-admission vital signs set not documented: 0T, 0P, 0R, 0S |
| 1 to 3 | Monitoring 1 to 3 times | Vital signs set documented 1 to 3 times |
| 4 to 7 | Monitoring 4 to 7 times | Vital signs set documented 4 to 7 times |
| 8 and above | Monitoring 8 times or more | Vital signs set documented 8 or more times |

The vital signs set is a complete set of TPRS

hypothesised impact model assumes both immediate (level) and month-to-month (slope) changes following the implementation of the intervention. To account for and quantify across-hospital variability, a random intercept term at the hospital level was included.

While designing the CNM charts, nurses in participating hospitals indicated that they typically monitor sicker babies more frequently [12]. For this reason, we include the Score for Essential Neonatal Symptoms and Signs (SENSS) (expressed in logits) in the ITS models as an explanatory variable. The SENNS score allows us to account for the severity of illness in the model and uses basic clinical signs to predict in-hospital mortality following neonatal unit admission in a low-resource, high-mortality setting [23]. There was no missing data in the outcome variables and imputation was done when calculating the SENSS score [24,25]. As the NEST360 program explicitly supported the introduction of pulse oximeters, we also include being a NEST360 site as an explanatory time-based covariate at the hospital level in our model. All analyses were performed using R Version 4.2.1 (R Foundation for Statistical Computing, Vienna, Austria; http://www.cran.r-project.org). We fitted an interrupted time series model of the form:

$$logit(P(y_i \leq j)) = \theta_j - \beta_1(intervention_i) - \beta_2(time_i) - \beta_3(trend_i) - \beta_4(SENSSscore_i) -$$
$$\beta_5(\mathrm{in}NESTProgram_i) - u(hospital_i)$$
$$j = 1, 2, \ldots, J - 1$$
$$i = 1, 2, \ldots, n$$

where $y_i$ is an ordinal outcome with $J$ categories, $\theta_j$ are threshold parameters of an ordinal logistic regression model, $\beta_1$ is the parameter for level change, $\beta_2$ is the parameter for slope before intervention, $\beta_3$ is the change in slope after intervention, $\beta_4$ captures the effect of SENSS score on the outcome, $\beta_5$ is the effect of a patient being admitted in a hospital enrolled in the nest program and $u$ are the random effects for hospitals. The "trend" variable indicates the number of months since intervention but takes the value zero for observations in the pre-intervention period.

We also tested whether the effect of introducing monitoring charts varied by severity of illness (SENSS score) by including interaction terms between SENSS score and variables for step change and slope change.

$$logit(P(y_i \leq j)) = \theta_j - \beta_1(intervention_i) - \beta_2(time_i) - \beta_3(trend_i) - \beta_4(SENSSscore_i) - \beta_5(\mathrm{in}NESTProgram_i) -$$
$$\beta_6(intervention_i xSENSSscore_i) - \beta_7(trend_i xSENSSscore_i) - u(hospital_i)$$
$$j = 1, 2, \ldots, J - 1$$
$$i = 1, 2, \ldots, n$$

where the parameters $\beta_6$ and $\beta_7$ estimated the interaction between severity of illness and both step change and slope change. SENSS score was scaled by subtracting the mean and dividing by the standard deviation to enhance model convergence. To enhance interpretability of interaction terms involving a continuous variable for SENSS score, we plotted the odds ratios of both step change and slope change against SENSS score.

**Audit of filled monitoring charts.** To complement the quantitative evaluation, an exploratory review of a sample of hospitals' filled CNM charts was undertaken. All data clerks in the 19 hospitals were requested to submit anonymised, scanned charts to allow us to describe how they were filled and identify opportunities for further improving the chart design. The data clerks submitted anonymised filled monitoring charts for review; 230 files (including any monitoring charts, treatment sheets and admission form) were submitted in January 2021

while 55 files (monitoring charts only) were received in 2022. Out of these, a maximum of the first four filled CNM monitoring charts were reviewed per hospital for a total of 86 reviewed charts (range 3–4 per hospital) from 13 hospitals. The review process involved checking if a CNM chart was present and used then noting how the three major sections were filled–vital signs and assessment, feed and fluid prescription and input balancing section.

## Ethics

The study uses de-identified data from the Clinical Information Network. Ethical approval for this study was granted by the Kenya Medical Research Institute (KEMRI) Scientific and Ethics Review Unit (Protocol no: KEMRI/SERU/CGMRC/161/3852).

## Results

### Sample characteristics

The study sample was 43,719 newborns across the 19 hospitals, with median monthly admissions per hospital ranging from 20 to 280 patients per hospital, and a median length of stay of 5 (Interquartile range of 3–10) days (S1 Appendix). During the 24-month study period, the median birth weight of newborn unit babies was 3.0kg (Interquartile range: 2.0–3.3) and 55.5% were male. The four most common primary diagnoses on admission were low birth weight (22.2%), birth asphyxia (19.4%), respiratory distress syndrome (18.1%), and neonatal sepsis (13.8%). The in-patient mortality among this sample of newborns admitted on the day of their birth and who survived up to 48 hours after admission was 6.6% (2,864/43,719).

### Descriptive analysis

Overall, the percentage of patients in the no-monitoring category decreased from 68.5% in the pre-intervention period to 43.5% in the post-intervention period for TPRS monitoring. In all subsequent categories, the level of monitoring increased (Table 3). The 1–3 times monitoring category recorded the highest increase in documentation (from 11.4% to 22.7%). A similar improvement pattern was observed if the categorical outcome was redefined to exclude S, which might be affected by the availability of pulse oximeters although nurses are encouraged to record if the baby is cyanosed (or not) in the absence of a pulse oximeter. The percentage of babies in the no monitoring category based just on TPR declined from 54.8% to 33.1% (Table 1 in S2 Appendix).

**Main outcome–TPRS composite outcome.** Fig 2 shows the changing pattern for categories of TPRS documentation per month; an improvement over time is seen when data for all 19 hospitals are combined, including the 3 EA hospitals (Panel A). At the end of the intervention period (October 2021), data from all 19 hospitals show a very similar pattern to analyses restricted to 16 non-EA hospitals (Panel B). Across the pre- and post-intervention period, it is apparent that approximately half of all newborns in the sample did not have a single full set of TPRS vital signs documented in during the first 48 hours.

**Table 3. Summary T-P-R-S monitoring descriptive analysis (N = 43,719 newborns from 19 hospitals).**

| TPRS Monitoring category | Before, N = 22,023 n (%) | After, N = 21,696 n (%) |
|---|---|---|
| No Monitoring | 15,089 (68.5%) | 9,434 (43.5%) |
| Monitoring 1 to 3 times | 2,503 (11.4%) | 4,915 (22.7%) |
| Monitoring 4 to 7 times | 1,669 (7.6%) | 3,573 (16.5%) |
| Monitoring 8 times or more | 2,762 (12.5%) | 3,774 (17.4%) |

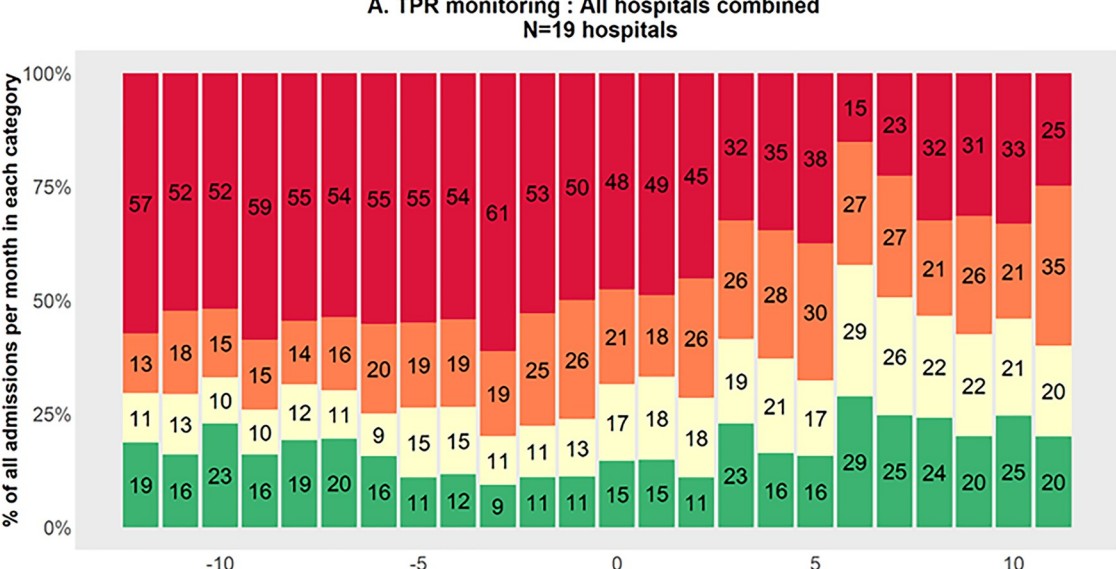

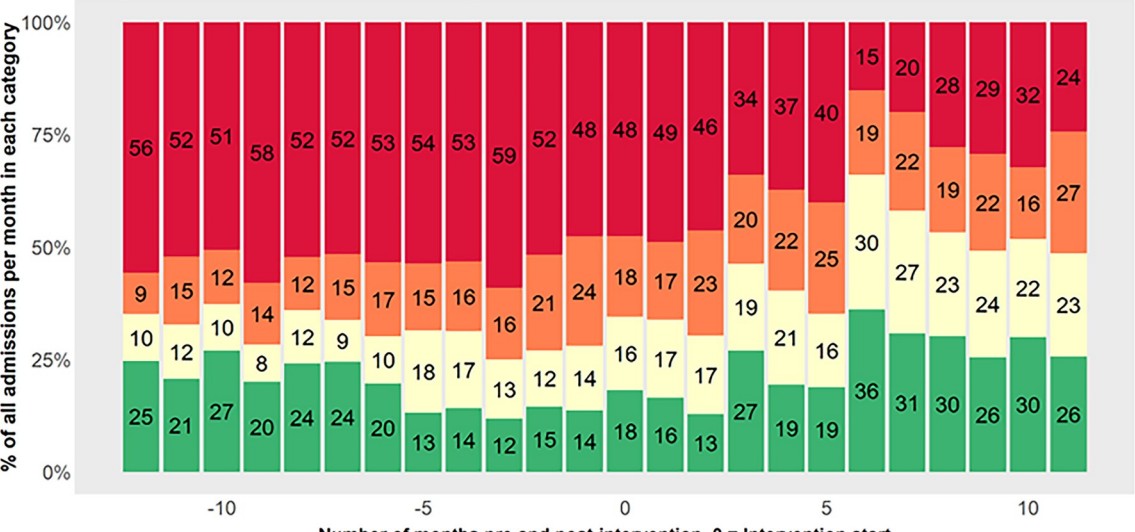

**Fig 2. Percentage of patients' TPRS monitored in each category over time (month 0 –intervention start).**

The individual hospital-level analysis showed heterogenous performance as illustrated in Fig 1 in S2 Appendix. H11 was the earliest adopter hospital (in May 2019, a full year before the official launch), with most of the patients having 8 or more TPRS monitoring set documented in the 24-month study period. Eight hospitals (H1, H6, H7, H9, H12, H14, H17 and H18) showed an improvement in the post-intervention period. On the other hand, H16 recorded no improvement in the pre- and post-intervention periods. When the results were analysed

excluding Oxygen Saturation(S)–so based on the TPR categorical outcome—H16 showed better performance in the post-intervention period.

**Explanatory analysis–individual vital signs.** Among all 19 hospitals the percentage of babies who received at least one reading each of T, P, R and S showed an increasing trend. Documentation of Cyanosis or measured oxygen saturation (S) began from the lowest baseline (only 31% of babies had at least 1 observation documented) and recorded the highest improvement (with 67% of babies having at least one observation documented, Fig 3 –panel A). A similar improvement pattern was observed when the 3 EA hospitals were excluded (Fig 3 –panel B). The improvement over time in the other three vital signs (TPR) showed a similar pattern.

Similar to the categorical outcome, individual vital signs analysis showed a heterogeneous performance across hospitals (Fig 2 in S2 Appendix). EA (H10, H11 and H15) and two other hospitals (H3 and H18) showed a consistent documentation pattern of at least one post-admission vital sign documented for a majority of the newborns in both the pre- and post-intervention periods. Eight hospitals (H1, H2, H6, H7, H9, H12, H14 and H17) showed an improvement in documentation of at least one vital sign after admission in the post-intervention period.

## Interrupted time series analysis

A total of 34,986 patients were included from 16 hospitals in the ITS analysis (three EA hospitals were excluded).

**Main outcome–TPRS composite outcome.** The intervention increased the odds of being in a higher TPRS monitoring category by 4.8 times (p<0.001) (Table 4). In the pre-intervention period, the odds of vital signs being documented in a higher monitoring category were declining by 4% (p <0.001) every month. There was 16% (p <0.001) change in trend post intervention implying that the odds of being in a higher monitoring category during the post intervention period was increasing by 11% (0.96 x 1.16). With every unit increase in the SENSS score (representing increasing severity of illness, expressed in logits), the odds of the vital signs being documented in a higher monitoring category increased by 14% (p<0.001). Being part of the NEST360 program increased the odds of the vital signs being documented in a higher monitoring category by 2.25 times (p<0.001).

**Explanatory analyses–individual vital signs.** For each of the individual vital signs, the odds of being documented in a higher monitoring category was declining in the pre-intervention period every month while the change in odds of being in a higher monitoring category increased by 17–22% per month (change in slope) in the post-intervention period. The intervention increased the odds of being in a higher monitoring category for each vital sign as shown in Table 5. Notably, the odds of oxygen saturation/cyanosis (S) being documented in a higher monitoring category were more than 5 times higher (OR = 5.60, p<0.001) in the post-intervention period. Additionally, being in the NEST360 program increased the odds of each vital sign being documented in a higher monitoring category by twofold for T, R and P and 89% for S.

**Effect modification by severity of illness (interaction).** There was statistical evidence that the effect of monitoring charts on TPRS, respiratory rate, oxygen saturation, and temperature monitoring was modified by severity of illness (the model for pulse rate did not converge). For the four models that converged, the odds ratios for step change decreased with increase in SENSS score, while the odds ratios of slope change increased with increase in SENSS score (Table 1 in S4 Appendix). However, a plot of odds ratios for step change and slope change against SENSS score showed that the difference in odds ratios between the sickest children and the less sick was not large. For TPRS, the odds ratios of step changed ranged between 1.50 and 1.75, while those of slope change ranged between 1.12 and 1.16 (Fig 1 in S4 Appendix).

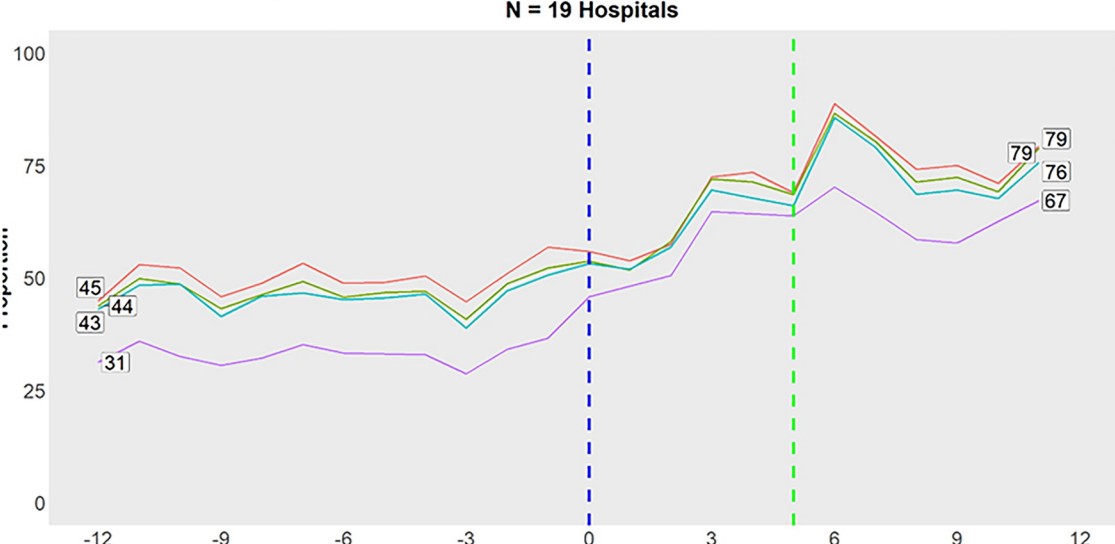

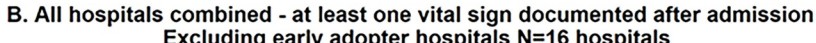

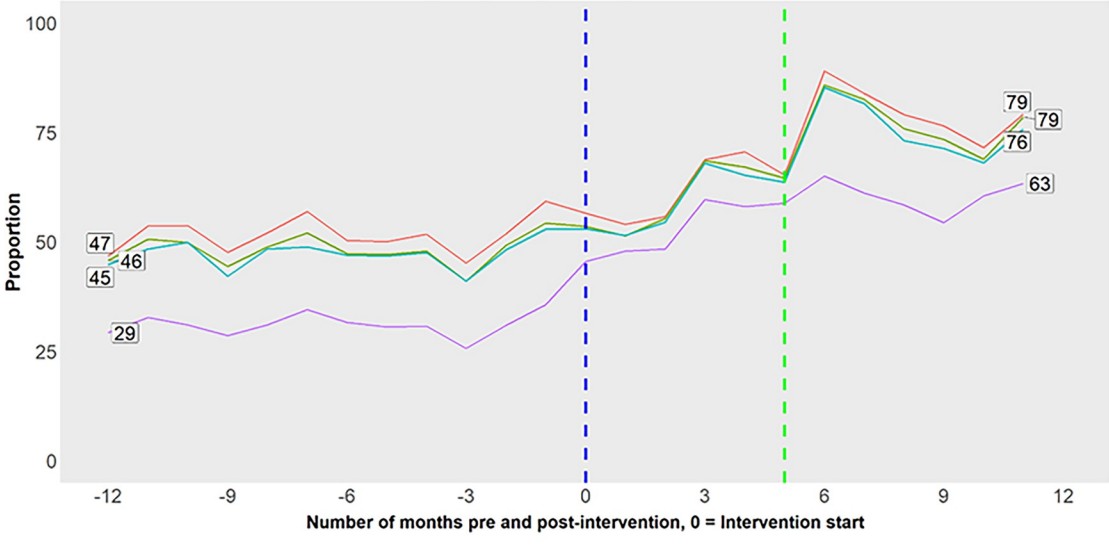

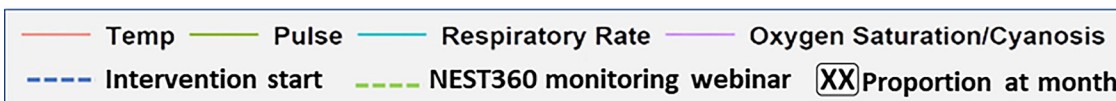

**Fig 3. Individual vital signs over time for all 19(all hospitals) versus 16(excluding early adopters) hospitals.**

## Audit of monitoring charts

A total of 86 comprehensive newborn monitoring charts from 13 hospitals were reviewed (range 3–4 per hospital). A descriptive summary of observations is presented with illustrations provided in the appendices (S3 Appendix). In some instances where the patient record had both CNM and old monitoring charts, the CNM charts were used while the old charts

**Table 4. TPRS monitoring outcome category–ITS analysis (in 16 non-EA hospitals, n = 34,986).**

| | TPRS Categorical Outcome | |
|---|---|---|
| | aOR (95% CI) | P-value |
| **Intervention** | 4.82 (4.24–5.48) | <0.001 |
| **Pre-intervention trend** | 0.96 (0.95–0.97) | <0.001 |
| **Change in trend** | 1.16 (1.14–1.19) | <0.001 |
| **SENSS score** | 1.14 (1.11–1.17) | <0.001 |
| **In NEST Program** | 2.25 (2.02–2.51) | <0.001 |

aOR: Adjusted Odds Ratio, CI: 95% Confidence Interval.

remained unused. In one hospital, the CNM chart was available and unused, but monitoring details were recorded in 'free text' nursing notes (the Kardex).

The vital signs and assessment section is the largest section on the CNM containing 12 columns, allowing for up to 2-hourly monitoring on one face of an A4 sheet in 24 hours. From our observation, many charts had one or two columns filled, while others had the whole section filled (all columns and rows). For charts that had the entire section filled, the charts were used for more than one day with users opting to add dates and timing at the top to denote a new day. In some instances, extra columns were added by hand at the extreme right-hand side of the chart to document vital signs monitoring and in one case, we found a hand-drawn chart that had columns similar to the CNM. Only one of the 86 charts was well filled with the different sections fully completed as per the intended use.

## Discussion

This study aimed to evaluate the recording of vital signs (temperature, pulse, respiratory rate, and oxygen saturation) after implementation of the newly designed CNM chart. The chart was co-designed with participants from the CIN-N hospitals, piloted in four hospitals between March and August 2019 and then launched in 19 hospitals virtually in July 2020 [12].

Overall, there was increased documentation for all vital signs in both the categorical grouped outcome and individual vital signs documentation. This finding is in line with literature, where better documentation was observed following the improvement of charts in various settings [26]. At the start of the study period, about 50% of babies did not get a complete TPRS set in the admission episode. At the end of the intervention period, the percentage reduced to 37% for all hospitals combined, implying that the intervention positively influenced the documentation of vital signs, but there remains room for improvement. Poor nurse-to-baby ratios may have contributed to observed documentation outcomes–a challenge that was identified throughout the

**Table 5. Individual vital signs ITS analysis (in 16 non-EA hospitals, n = 34,986).**

| | Temperature | | Pulse | | Respiratory Rate | | Oxygen Saturation/Cyanosis | |
|---|---|---|---|---|---|---|---|---|
| | aOR (CI) | P-value | aOR (CI) | P-value | aOR (CI) | P-value | aOR (95% CI) | P-value |
| Intervention | 1.59 (1.44–1.75) | <0.001 | 1.76 (1.59–1.94) | <0.001 | 1.77 (1.60–1.96) | <0.001 | 5.60 (4.94–6.35) | <0.001 |
| Pre-intervention trend | 0.93 (0.92–0.94) | <0.001 | 0.94 (0.93–0.95) | <0.001 | 0.96 (0.95–0.96) | <0.001 | 0.94 (0.92–0.95) | <0.001 |
| Change in trend | 1.20 (1.18–1.22) | <0.001 | 1.20 (1.18–1.22) | <0.001 | 1.17 (1.15–1.19) | <0.001 | 1.22 (1.19–1.24) | <0.001 |
| SENSS score | 1.14 (1.12–1.17) | <0.001 | 1.15 (1.13–1.18) | <0.001 | 1.15 (1.13–1.18) | <0.001 | 1.13 (1.11–1.16) | <0.001 |
| In NEST Program | 2.28 (2.09–2.49) | <0.001 | 2.09 (1.91–2.29) | <0.001 | 2.20 (2.01–2.41) | <0.001 | 1.89 (1.70–2.10) | <0.001 |

aOR: Adjusted Odds Ratio, CI: 95% Confidence Interval.

design and piloting period [12]. Similarly, previous work has shown that there are very poor nurse-to-baby ratios in newborn units of public hospitals (median ratio of 19 babies per nurse) contributing to missed nursing care, including documentation [27].

Following discussions with nurses that indicated that sicker babies got more monitoring during the chart design phase and the implementation phase [13], the SENNS score was included in the ITS models. In all the models, this was statistically significant, showing that there was an association between poor health and more frequent monitoring. This is consistent with the newborn care standards proposal that sicker babies receive more frequent monitoring than more stable babies [11,28,29]. Improving inpatient monitoring and its documentation contributes toward improving patient outcomes and reducing newborn mortality, which remains high in low- and middle-income countries. In this study, the CNM charts seemed to have fulfilled their intended use among the sickest babies.

The purpose of co-design was to overcome some well-documented problems with monitoring charts such as duplication of efforts [23,24], fragmentation [7,25] or improvisation of documentation by health workers such as the use of 'scraps' (of paper) for handover [30]. Additionally, standardised observation charts for monitoring care are recommended to facilitate documentation of care alongside other interventions [11,31,32]. However, it was evident from the chart review that duplication and improvisation were still occurring and that there was considerable variation in the use of the charts. The CNM chart provides the opportunity to record 2 hourly observations or up to 12 sets of observations in 24 hours, but this was rare from the observed charts. Some facilities were observed to use one face of the chart over several days rather than the anticipated 24-hour period. This may have reflected limited resources leading to staff filling all the blank spaces before documenting on additional sheets. Anecdotally we are aware that supplies of hospital stationery for medical records remain hard to sustain in some hospitals. Additionally, the chart review revealed that some old monitoring charts were still available in the patient file alongside the CNM charts leading to partial filling of the CNM charts or the CNM charts not being filled at all. Documentation was observed to be infrequent and erratic thereby providing insights into the poor documentation outcomes in the quantitative analysis.

For each vital sign, the odds of being in a higher monitoring category based on documentation increased with oxygen saturation/cyanosis assessment having the lowest baseline and recording the highest odds of improvement. Specifically, for hospitals in the NEST360 program, the training on comprehensive newborn care and provision of equipment which included pulse oximeters to measure oxygen saturation appeared to have a positive impact on documentation. The NEST360 program trained staff in the recruited hospitals on comprehensive newborn care including documentation. Additionally, they conducted a teaching webinar on newborn monitoring and documentation during the COVID-19 period which was open to all health workers in the country. During the webinar, anonymised documentation data were presented and following the webinar, informal discussions with staff at the hospitals showed that paediatricians showed more interest in their performance in documenting vital signs. This was evidenced by the slight increase in documentation noted at month 6 post-intervention when the NEST360 webinar was conducted. The NEST360 program was a complementary project to the chart launch as it provided additional training on documentation practices as well as the care of patients demonstrating that multifaceted interventions are required to achieve quality improvement.

Implementing a co-designed chart is not sufficient to ensure immediate and sustained chart uptake and more effort is required over an extended period to realise the benefits of the chart and reach recommended documentation standards. This was evident from the performance of early adopter hospitals in both TPRS composite analysis and individual vital signs

analysis. These early adopter hospitals implemented the chart earlier than the planned launch date of July 2020 through the initiative of the senior nurse or paediatrician, possibly because they were involved in the chart design phase. Moreover, there may be hospital-level factors that affected documentation as evidenced by the heterogeneous performance across the hospitals. One hospital (H20) recorded sustained, good documentation over the entire study period, and this was attributed to a proactive leadership that actively reached out to the NEST360 co-ordinator even though they were not part of the NEST360 program. These findings on heterogeneous hospital performance imply that, hospital-specific data may be used to prepare targeted training materials and support specific mentorship on documentation.

An earlier study conducted in a similar set of hospitals in Kenya assessed the implementation of a structured paediatric admission record (PAR) and reported a large improvement in its uptake and improvement in documentation of prescribed drugs and clinical signs on admission–a single time-point task (8). In contrast, our study assessed the documentation of vital signs which are collected repeatedly and regularly after admission. Arguably, this is just a snapshot of the tasks carried out by the nurses and, in prior our interactions with hospital staff during CIN-N meetings, nurses acknowledged poor monitoring owing to the large number of tasks they must do [33]. Using a human-centered approach to design enabled us to glean insights to interpret the quantitative analysis of vital signs documentation. For example, during the code-sign sessions, the nurses reported to frequently lumping tasks together to ease work processes and this is evident from the TPR correlation confirming that these signs are often viewed as a set and therefore done together. In another example, the nurses reported that they faced many challenges that hinder proper documentation such as limited staffing and lack of equipment such as lack of pulse oximeters to aid in documenting oxygen saturation and this was clear as oxygen saturation (S) was documented less frequently than other vital signs in the study period. Considering these insights, ensuring that many tasks, including documentation, are done consistently and according to standards will require that staffing issues are addressed [27,33].

**Limitations.** Documentation of vital signs may have been captured in other forms in the patient record that were not captured by the clerks using the standard operating procedures we employed; this means we may have underestimated to some degree the documentation of vital signs. While we set a launch date of July 2020, we learned that some hospitals opted to complete their supply of old monitoring charts and therefore actual CNM implementation dates may have been variable. However, there was agreement from all hospitals to use the CNM chart as the preferred and most important place to record vital signs. Further, during implementation, some nurses also reported adopting a phased approach to implementation where sicker babies were given priority and their documentation was done using the CNM charts [13]. Therefore, intervention effects may have been influenced by behavioural practices of health workers who may have opted to document vital signs elsewhere and hospitals that may have had a delayed start date. Additionally, the chart was launched at the height of the COVID-19 pandemic, and this may have affected the hospital's resources such as the ability to provide an adequate supply of charts and staff being re-deployed to isolation wards. A lack of adequate supply of charts and other resources may have limited the ability of staff to document care at the required frequency.

In this study, only babies with a minimum length of stay of 48 hours after admission were included. This may have introduced a selection bias as those patients may have been different from those who died within 24 hours of admission. The inclusion of the SENNS score as an explanatory variable in the models suggested better documentation for sicker babies which implies that the analysis may be underestimating the effect of the chart on documentation. Lastly, the hospitals included in the study received regular feedback on their performance

throughout the whole study period (including quality of documentation) and a subset of them were involved in the chart design phase. Therefore, the results may not be generalisable to other non-network hospitals in the country or other LMIC settings.

**Recommendations for future work.** In this study, we use documentation of vital signs as an indicator of chart adoption and did not examine the actual values documented, the subsequent care provided based on documented values, or the effect on survival. Such future work is needed to give insights into the quality of care and can be explored as the next level of analysis as a culture of quality and quantity documentation is established.

## Conclusion

The intervention increased the odds of vital signs being documented in a higher monitoring category for the categorical TPRS outcome as well as for each vital sign. However, none of the vital signs achieved universal coverage. While implementing a standardised co-designed monitoring chart had a positive effect on the documentation of key vital signs, that alone is not sufficient to ensure immediate and sustained chart uptake. More effort is required over an extended period in addition to solving fundamental issues like staffing ratios to realise the benefits of the chart and reach recommended documentation standards. Hospital-specific performance may be used to prepare targeted training materials and support ongoing mentorship on documentation, including by other co-implemented quality improvement projects which use the tool.

## Supporting information

**S1 Checklist. STROBE statement—checklist of items that should be included in reports of observational studies.**
(DOCX)

**S1 Appendix. Hospital characteristics.**
(DOCX)

**S2 Appendix. Detailed tables and charts.**
(DOCX)

**S3 Appendix. Chart review sample images.**
(DOCX)

**S4 Appendix. Interaction model results.**
(DOCX)

**S1 Text. PLOS questionnaire on inclusivity in global research.**
(DOCX)

## Acknowledgments

The CIN-N team for their support to hospitals during the implementation period and regular team meetings that also served as debriefing sessions. This work is published with the permission of the Director of KEMRI.

## Author Contributions

**Conceptualization:** Naomi Muinga, Chris Paton, Mike English.

**Data curation:** Timothy Tuti, Paul Mwaniki.

**Formal analysis:** Naomi Muinga, Timothy Tuti, Paul Mwaniki.

**Funding acquisition:** Mike English.

**Methodology:** Mike English.

**Supervision:** Lenka Beňová, Mike English.

**Visualization:** Naomi Muinga, Paul Mwaniki.

**Writing – original draft:** Naomi Muinga, Timothy Tuti, Paul Mwaniki, Edith Gicheha, Chris Paton, Lenka Beňová.

**Writing – review & editing:** Naomi Muinga, Timothy Tuti, Paul Mwaniki, Edith Gicheha, Chris Paton, Lenka Beňová, Mike English.

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
