## [Decision Letter · Decision Letter 0]

26 Jul 2023

PGPH-D-23-00617

Evaluating the documentation of vital signs following implementation of a new comprehensive newborn monitoring chart in 19 hospitals in Kenya: a time series analysis.

Dear Dr. Muinga,

Thank you for submitting your manuscript to PLOS Global Public Health. After careful consideration, we feel that it has merit but does not fully meet PLOS Global Public Health’s publication criteria as it currently stands. Therefore, we invite you to submit a revised version of the manuscript that addresses the points raised during the review process.

We look forward to receiving your revised manuscript.

Kind regards,

Abraham D. Flaxman, Ph.D.

Academic Editor

Journal Requirements:

Additional Editor Comments (if provided):

Reviewers' comments:

Reviewer's Responses to Questions

**Comments to the Author**

1. Does this manuscript meet PLOS Global Public Health’s publication criteria? Is the manuscript technically sound, and do the data support the conclusions? The manuscript must describe methodologically and ethically rigorous research with conclusions that are appropriately drawn based on the data presented.

Reviewer #1: Yes

Reviewer #2: Yes

2. Has the statistical analysis been performed appropriately and rigorously?

Reviewer #1: I don't know

Reviewer #2: Yes

3. Have the authors made all data underlying the findings in their manuscript fully available (please refer to the Data Availability Statement at the start of the manuscript PDF file)?

Reviewer #1: No

Reviewer #2: Yes

4. Is the manuscript presented in an intelligible fashion and written in standard English?

Reviewer #1: Yes

Reviewer #2: Yes

5. Review Comments to the Author

Reviewer #1: Thank you for the opportunity to review this interesting paper! I really like this paper — it addresses an important gap in health care quality, gives a lot of rich contextual information about the study setting, describes an interesting intervention, and includes very rich data on the effects of the intervention. I have several questions and suggestions for strengthening the paper, mainly focused on the interrupted time-series analysis. Please see below:

Comments/questions on the intervention:

1. Did the intervention include any quality-checks on the data reported on the charts? Is there a way to know whether the charts were actually used as intended, vs. completed at the end of the day by a nurse? (If no - perhaps future work could include some direct observation of care to document whether charts are used as intended).

2. Do you have any information about whether the intervention led to increased monitoring of vital signs, as opposed to only increased documentation of monitoring? In some places in the manuscript you talk about ‘higher monitoring’ but in others you talk mainly about better documentation. It would be interesting to understand more about whether the charts actually encourage providers to check vital signs more frequently vs. giving them a clearer/more organized way to record vital signs information.

Comments/questions on the interrupted time-series analysis:

3. To better understand the analysis and the interpretation of the results, it would be helpful if you could include a regression equation in the manuscript.

4. The authors chose to use a logistic regression, but this model could have also potentially been fit as a linear probability model, which would have been easier to interpret. The logistic regression assumes that the trend over time in vital signs documentation is linear in the log odds…did you find this to be the case? It would be helpful to include a justification for the use of logistic regression here, and to include results for a linear probability model as a sensitivity analysis.

5. I found some of the phrasing of the results to be fairly confusing. For example, “In the pre-intervention period, the odds of being in a higher monitoring category were declining by 4% every month while the change in the odds of being in a higher monitoring category increased by 16 per month in the post-intervention period.” If I understand the model correctly, I think you could instead write this as, “In the pre-intervention period, the odds of being in a higher monitoring category were declining by 4% every month. Following the intervention, there was a change in this trend: the odds began to increase by 12% per month.” (This is assuming that the model is something like y = a + b1*month + b2*post + b3*month*post, and b1=-0.04, and b2=0.16.) There are several places where I feel that the language describing the results could be clarified in this way.

6. Figure 2- why does it look like vital signs documentation decreased after the webinar?

7. In the analysis, you adjust for how sick the baby is. By including this variable in the model, you essentially allow for a higher intercept (more vital signs documentation) for the sicker babies. However, I think it would be interesting to know whether the intervention actually worked better (or worse) for these babies. To do this, you could include an interaction between the “post” variable for the intervention and the sickness variable.

Other comments:

8. In the introduction, it would be useful to have an additional 1-2 sentences describing why documentation of vital signs is so important in neonatal wards. (You reference previous work, which explains this well, but I think that some readers may miss this and it could be stated more clearly upfront in this manuscript).

Reviewer #2: I appreciate the opportunity to review this manuscript. The authors' effort to improve the quality of care for sick newborns through more frequent monitoring of vital signs is highly commendable.

The study was done according to high methodological standards. I see no significant methodological or formal errors in the manuscript. It is well-written and has lots of essential information. It was easy to follow with a good explanation of the study rationale, a reference to the design process of the monitoring chart used, a clear presentation of results, and a discussion of findings leading to a sound conclusion.

Overall, the study has value, and I will recommend it for publication. I have some minor queries you might find helpful in progressing your study report toward publication.

The description of the study design could have been more explicit. Were there any qualitative scoping activities (i.e. interviews or FGDs)? These were not mentioned in the methodology or the result sections.

I note some quotes from nurses in the last paragraph of the discussion section. Are these from KIIs/FGDs? Please clarify.

Considering the above, it may be worth revisiting your description of the study design- description of methods. Should it be quantitative rather than a mixed design?

The same reason why I found the use of the term "Qualitative audit of monitoring charts" to describe a review of the quality of documentation of vital signs a misnomer. Please clarify

6. PLOS authors have the option to publish the peer review history of their article (what does this mean?). If published, this will include your full peer review and any attached files.

**Do you want your identity to be public for this peer review?** For information about this choice, including consent withdrawal, please see our Privacy Policy.

Reviewer #1: No

Reviewer #2: **Yes: **Dr Aminu Umar

---

## [Decision Letter · Decision Letter 1]

5 Oct 2023

Evaluating the documentation of vital signs following implementation of a new comprehensive newborn monitoring chart in 19 hospitals in Kenya: a time series analysis.

PGPH-D-23-00617R1

Dear Ms Muinga,

We are pleased to inform you that your manuscript 'Evaluating the documentation of vital signs following implementation of a new comprehensive newborn monitoring chart in 19 hospitals in Kenya: a time series analysis.' has been provisionally accepted for publication in PLOS Global Public Health.

Best regards,

Abraham D. Flaxman, Ph.D.

Academic Editor

Reviewer Comments (if any, and for reference):

Reviewer's Responses to Questions

**Comments to the Author**

1. If the authors have adequately addressed your comments raised in a previous round of review and you feel that this manuscript is now acceptable for publication, you may indicate that here to bypass the “Comments to the Author” section, enter your conflict of interest statement in the “Confidential to Editor” section, and submit your "Accept" recommendation.

Reviewer #1: (No Response)

Reviewer #2: All comments have been addressed

2. Does this manuscript meet PLOS Global Public Health’s publication criteria? Is the manuscript technically sound, and do the data support the conclusions? The manuscript must describe methodologically and ethically rigorous research with conclusions that are appropriately drawn based on the data presented.

Reviewer #1: Yes

Reviewer #2: Yes

3. Has the statistical analysis been performed appropriately and rigorously?

Reviewer #1: Yes

Reviewer #2: Yes

4. Have the authors made all data underlying the findings in their manuscript fully available (please refer to the Data Availability Statement at the start of the manuscript PDF file)?

Reviewer #1: Yes

Reviewer #2: Yes

5. Is the manuscript presented in an intelligible fashion and written in standard English?

Reviewer #1: Yes

Reviewer #2: Yes

6. Review Comments to the Author

Reviewer #1: Thank you for the opportunity to review the revised manuscript. The authors have addressed all of my comments. I just have a couple of minor suggestions for the presentation of the regression model:

1) I think there is a typo in the model - there should be plus signs (+) between the various coefficients rather than minus signs (-)

2) In addition, I would recommend renaming "Time" to "Trend," and renaming "Trend" to "Change in trend." It seems clearer to me to use the same word ("trend") for these to indicate that one of them is the baseline value and one is the change in this value after the intervention.

Congrats on a great manuscript!

Reviewer #2: My queries have been addressed satisfactorily. I have no further questions.

7. PLOS authors have the option to publish the peer review history of their article (what does this mean?). If published, this will include your full peer review and any attached files.

**Do you want your identity to be public for this peer review?** For information about this choice, including consent withdrawal, please see our Privacy Policy.

Reviewer #1: No

Reviewer #2: **Yes: **Dr Aminu Umar
